# PROXIMAL BACKPROPAGATION

**Thomas Frerix**[1][*]**, Thomas Möllenhoff**[1][*]**, Michael Moeller**[2][*]**, Daniel Cremers**[1]

```
thomas.frerix@tum.de
thomas.moellenhoff@tum.de
michael.moeller@uni-siegen.de
cremers@tum.de
```

[1] Technical University of Munich
[2] University of Siegen

## ABSTRACT

We propose proximal backpropagation (ProxProp) as a novel algorithm that takes *implicit* instead of explicit gradient steps to update the network parameters during neural network training. Our algorithm is motivated by the step size limitation of explicit gradient descent, which poses an impediment for optimization. Prox-Prop is developed from a general point of view on the backpropagation algorithm, currently the most common technique to train neural networks via stochastic gradient descent and variants thereof. Specifically, we show that backpropagation of a prediction error is equivalent to sequential gradient descent steps on a quadratic penalty energy, which comprises the network activations as variables of the optimization. We further analyze theoretical properties of ProxProp and in particular prove that the algorithm yields a descent direction in parameter space and can therefore be combined with a wide variety of convergent algorithms. Finally, we devise an efficient numerical implementation that integrates well with popular deep learning frameworks. We conclude by demonstrating promising numerical results and show that ProxProp can be effectively combined with common first order optimizers such as Adam.

## 1 INTRODUCTION

In recent years neural networks have gained considerable attention in solving difficult correlation tasks such as classification in computer vision (Krizhevsky et al., 2012) or sequence learning (Sutskever et al., 2014) and as building blocks of larger learning systems (Silver et al., 2016). Training neural networks is accomplished by optimizing a nonconvex, possibly nonsmooth, nested function of the network parameters. Since the introduction of stochastic gradient descent (SGD) (Robbins & Monro, 1951; Bottou, 1991), several more sophisticated optimization methods have been studied. One such class is that of quasi-Newton methods, as for example the comparison of L-BFGS with SGD in (Le et al., 2011), Hessian-free approaches (Martens, 2010), and the Sum of Functions Optimizer in (Sohl-Dickstein et al., 2013). Several works consider specific properties of energy landscapes of deep learning models such as frequent saddle points (Dauphin et al., 2014) and well-generalizable local optima (Chaudhari et al., 2017a). Among the most popular optimization methods in currently used deep learning frameworks are momentum based improvements of classical SGD, notably Nesterov's Accelerated Gradient (Nesterov, 1983; Sutskever et al., 2013), and the Adam optimizer (Kingma & Ba, 2015), which uses estimates of first and second order moments of the gradients for parameter updates.

Nevertheless, the optimization of these models remains challenging, as learning with SGD and its variants requires careful weight initialization and a sufficiently small learning rate in order to yield a stable and convergent algorithm. Moreover, SGD often has difficulties in propagating a learning signal deeply into a network, commonly referred to as the vanishing gradient problem (Hochreiter et al., 2001).

---

[*]contributed equally

Figure 1: Notation overview. For an $L$-layer feed-forward network we denote the explicit layer-wise activation variables as $z_l$ and $a_l$. The transfer functions are denoted as $\phi$ and $\sigma$. Layer $l$ is of size $n_l$.

Training neural networks can be formulated as a constrained optimization problem by explicitly introducing the network activations as variables of the optimization, which are coupled via layer-wise constraints to enforce a feasible network configuration. The authors of (Carreira-Perpiñán & Wang, 2014) have tackled this problem with a quadratic penalty approach, the method of auxiliary coordinates (MAC). Closely related, (Taylor et al., 2016) introduce additional auxiliary variables to further split linear and nonlinear transfer between layers and propose a primal dual algorithm for optimization. From a different perspective, (LeCun, 1988) takes a Lagrangian approach to formulate the constrained optimization problem.

In this work, we start from a constrained optimization point of view on the classical backpropagation algorithm. We show that backpropagation can be interpreted as a method alternating between two steps. First, a forward pass of the data with the current network weights. Secondly, an ordered sequence of gradient descent steps on a quadratic penalty energy.

Using this point of view, instead of taking explicit gradient steps to update the network parameters associated with the *linear* transfer functions, we propose to use implicit gradient steps (also known as proximal steps, for the definition see (6)). We prove that such a model yields a descent direction and can therefore be used in a wide variety of (provably convergent) algorithms under weak assumptions. Since an exact proximal step may be costly, we further consider a matrix-free conjugate gradient (CG) approximation, which can directly utilize the efficient pre-implemented forward and backward operations of any deep learning framework. We prove that this approximation still yields a descent direction and demonstrate the effectiveness of the proposed approach in PyTorch.

## 2 MODEL AND NOTATION

We propose a method to train a general $L$-layer neural network of the functional form

$$J(\boldsymbol{\theta}; X, y) = \mathcal{L}_y(\phi(\theta_{L-1}, \sigma(\phi(\cdots, \sigma(\phi(\theta_1, X))\cdots)))). \tag{1}$$

Here, $J(\boldsymbol{\theta}; X, y)$ denotes the training loss as a function of the network parameters $\boldsymbol{\theta}$, the input data $X$ and the training targets $y$. As the final loss function $\mathcal{L}_y$ we choose the softmax cross-entropy for our classification experiments. $\phi$ is a linear transfer function and $\sigma$ an elementwise nonlinear transfer function. As an example, for fully-connected neural networks $\theta = (W, b)$ and $\phi(\theta, a) = Wa + b\mathbf{1}$.

While we assume the nonlinearities $\sigma$ to be continuously differentiable functions for analysis purposes, our numerical experiments indicate that the proposed scheme extends to rectified linear units (ReLU), $\sigma(x) = \max(0, x)$. Formally, the functions $\sigma$ and $\phi$ map between spaces of different dimensions depending on the layer. However, to keep the presentation clean, we do not state this dependence explicitly. Figure 1 illustrates our notation for the fully-connected network architecture.

Throughout this paper, we denote the Euclidean norm for vectors and the Frobenius norm for matrices by $||\cdot||$, induced by an inner product $\langle\cdot, \cdot\rangle$. We use the gradient symbol $\nabla$ to denote the transpose of the Jacobian matrix, such that the chain rule applies in the form "inner derivative times outer derivative". For all computations involving matrix-valued functions and their gradient/Jacobian, we uniquely identify all involved quantities with their vectorized form by flattening matrices in a column-first order. Furthermore, we denote by $A^*$ the adjoint of a linear operator $A$.

| **Algorithm 1** - Penalty formulation of BackProp | **Algorithm 2** - ProxProp |
|---|---|
| **Input:** Current parameters $\boldsymbol{\theta}^k$. | **Input:** Current parameters $\boldsymbol{\theta}^k$. |
| *// Forward pass.*
**for** $l = 1$ **to** $L - 2$ **do**
$\quad z_l^k = \phi(\theta_l^k, a_{l-1}^k),$       *// $a_0 = X$.*
$\quad a_l^k = \sigma(z_l^k).$
**end for** | *// Forward pass.*
**for** $l = 1$ **to** $L - 2$ **do**
$\quad z_l^k = \phi(\theta_l^k, a_{l-1}^k),$       *// $a_0 = X$.*
$\quad a_l^k = \sigma(z_l^k).$
**end for** |
| *// Perform minimization steps on* (3).
ⓐ grad. step on $E$ wrt. $(\theta_{L-1}, a_{L-2})$
**for** $l = L - 2$ **to** $1$ **do**
$\quad$ⓑ grad. step on $E$ wrt. $z_l$ and $a_{l-1}$,
$\quad$ⓒ grad. step on $E$ wrt. $\theta_l$.
**end for** | *// Perform minimization steps on* (3).
ⓐ grad. step on $E$ wrt. $(\theta_{L-1}, a_{L-2})$, Eqs. 8, 12.
**for** $l = L - 2$ **to** $1$ **do**
$\quad$ⓑ grad. step on $E$ wrt. $z_l$ and $a_{l-1}$, Eqs. 9, 10.
$\quad$ⓒ prox step on $E$ wrt. $\theta_l$, Eq. 11.
**end for** |
| **Output:** New parameters $\boldsymbol{\theta}^{k+1}$. | **Output:** New parameters $\boldsymbol{\theta}^{k+1}$. |

## 3 PENALTY FORMULATION OF BACKPROPAGATION

The gradient descent iteration on a nested function $J(\boldsymbol{\theta}; X, y)$,

$$\boldsymbol{\theta}^{k+1} = \boldsymbol{\theta}^k - \tau \nabla J(\boldsymbol{\theta}^k; X, y), \tag{2}$$

is commonly implemented using the backpropagation algorithm (Rumelhart et al., 1986). As the basis for our proposed optimization method, we derive a connection between the classical backpropagation algorithm and quadratic penalty functions of the form

$$E(\boldsymbol{\theta}, \boldsymbol{a}, \boldsymbol{z}) = \mathcal{L}_y(\phi(\theta_{L-1}, a_{L-2})) + \sum_{l=1}^{L-2} \frac{\gamma}{2}\|\sigma(z_l) - a_l\|^2 + \frac{\rho}{2}\|\phi(\theta_l, a_{l-1}) - z_l\|^2. \tag{3}$$

The approach of (Carreira-Perpiñán & Wang, 2014) is based on the minimization of (3), as under mild conditions the limit $\rho, \gamma \to \infty$ leads to the convergence of the sequence of minimizers of $E$ to the minimizer of $J$ (Nocedal & Wright, 2006, Theorem 17.1). In contrast to (Carreira-Perpiñán & Wang, 2014) we do not optimize (3), but rather use a connection of (3) to the classical backpropagation algorithm to motivate a semi-implicit optimization algorithm for the original cost function $J$.

Indeed, the iteration shown in Algorithm 1 of forward passes followed by a sequential gradient descent on the penalty function $E$ is equivalent to the classical gradient descent iteration.

**Proposition 1.** *Let $\mathcal{L}_y$, $\phi$ and $\sigma$ be continuously differentiable. For $\rho = \gamma = 1/\tau$ and $\boldsymbol{\theta}^k$ as the input to Algorithm 1, its output $\boldsymbol{\theta}^{k+1}$ satisfies (2), i.e., Algorithm 1 computes one gradient descent iteration on $J$.*

*Proof.* For this and all further proofs we refer to Appendix A. □

## 4 PROXIMAL BACKPROPAGATION

The interpretation of Proposition 1 leads to the natural idea of replacing the explicit gradient steps ⓐ, ⓑ and ⓒ in Algorithm 1 with other – possibly more powerful – minimization steps. We propose Proximal Backpropagation (ProxProp) as one such algorithm that takes *implicit* instead of *explicit* gradient steps to update the network parameters $\theta$ in step ⓒ. This algorithm is motivated by the step size restriction of gradient descent.

### 4.1 GRADIENT DESCENT AND PROXIMAL MAPPINGS

Explicit gradient steps pose severe restrictions on the allowed step size $\tau$: Even for a convex, twice continuously differentiable, $\mathcal{L}$-smooth function $f : \mathbb{R}^n \to \mathbb{R}$, the convergence of the gradient

descent algorithm can only be guaranteed for step sizes $0 < \tau < 2/\mathscr{L}$. The Lipschitz constant $\mathscr{L}$ of the gradient $\nabla f$ is in this case equal to the largest eigenvalue of the Hessian $H$. With the interpretation of backpropagation as in Proposition 1, gradient steps are taken on quadratic functions. As an example for the first layer,

$$f(\theta) = \frac{1}{2}||\theta X - z_1||^2 . \tag{4}$$

In this case the Hessian is $H = X X^\top$, which is often ill-conditioned. For the CIFAR-10 dataset the largest eigenvalue is $6.7 \cdot 10^6$, which is seven orders of magnitude larger than the smallest eigenvalue. Similar problems also arise in other layers where poorly conditioned matrices $a_l$ pose limitations for guaranteeing the energy $E$ to decrease.

The proximal mapping (Moreau, 1965) of a function $f : \mathbb{R}^n \to \mathbb{R}$ is defined as:

$$\text{prox}_{\tau f}(y) := \underset{x \in \mathbb{R}^n}{\text{argmin}} \; f(x) + \frac{1}{2\tau}||x - y||^2. \tag{5}$$

By rearranging the optimality conditions to (5) and taking $y = x^k$, it can be interpreted as an *implicit* gradient step evaluated at the new point $x^{k+1}$ (assuming differentiability of $f$):

$$x^{k+1} := \underset{x \in \mathbb{R}^n}{\text{argmin}} \; f(x) + \frac{1}{2\tau}||x - x^k||^2 = x^k - \tau \nabla f(x^{k+1}). \tag{6}$$

The iterative algorithm (6) is known as the proximal point algorithm (Martinet, 1970). In contrast to explicit gradient descent this algorithm is *unconditionally stable*, i.e. the update scheme (6) monotonically decreases $f$ for any $\tau > 0$, since it holds by definition of the minimizer $x^{k+1}$ that $f(x^{k+1}) + \frac{1}{2\tau}||x^{k+1} - x^k||^2 \le f(x^k)$.

Thus proximal mappings yield unconditionally stable subproblems in the following sense: The update in $\theta_l$ provably decreases the penalty energy $E(\boldsymbol{\theta}, \boldsymbol{a}^k, \boldsymbol{z}^k)$ from (3) for fixed activations $(\boldsymbol{a}^k, \boldsymbol{z}^k)$ for any choice of step size. This motivates us to use proximal steps as depicted in Algorithm 2.

## 4.2 PROXPROP

We propose to replace explicit gradient steps with proximal steps to update the network parameters of the linear transfer function. More precisely, after the forward pass

$$\begin{aligned} z_l^k &= \phi(\theta_l^k, a_{l-1}^k), \\ a_l^k &= \sigma(z_l^k), \end{aligned} \tag{7}$$

we keep the explicit gradient update equations for $z_l$ and $a_l$. The last layer update is

$$a_{L-2}^{k+1/2} = a_{L-2}^k - \tau \nabla_{a_{L-2}} \mathcal{L}_y(\phi(\theta_{L-1}, a_{L-2})), \tag{8}$$

and for all other layers,

$$z_l^{k+1/2} = z_l^k - \sigma'(z_l^k)(\sigma(z_l^k) - a_l^{k+1/2}), \tag{9}$$

$$a_{l-1}^{k+1/2} = a_{l-1}^k - \nabla \left( \frac{1}{2}||\phi(\theta_l, \cdot) - z_l^{k+1/2}||^2 \right)(a_{l-1}^k), \tag{10}$$

where we use $a_l^{k+1/2}$ and $z_l^{k+1/2}$ to denote the updated variables before the forward pass of the next iteration and multiplication in (9) is componentwise. However, instead of taking explicit gradient steps to update the linear transfer parameters $\theta_l$, we take proximal steps

$$\theta_l^{k+1} = \underset{\theta}{\text{argmin}} \; \frac{1}{2}||\phi(\theta, a_{l-1}^k) - z_l^{k+1/2}||^2 + \frac{1}{2\tau_\theta}||\theta - \theta_l^k||^2. \tag{11}$$

This update can be computed in closed form as it amounts to a linear solve (for details see Appendix B). While in principle one can take a proximal step on the final loss $\mathcal{L}_y$, for efficiency reasons we choose an explicit gradient step, as the proximal step does not have a closed form solution in many scenarios (e.g. the softmax cross-entropy loss in classification problems). Specifically,

$$\theta_{L-1}^{k+1} = \theta_{L-1}^k - \tau \nabla_{\theta_{L-1}} \mathcal{L}_y(\phi(\theta_{L-1}^k, a_{L-2}^k)). \tag{12}$$

Note that we have eliminated the step sizes in the updates for $z_l$ and $a_{l-1}$ in (9) and (10), as such updates correspond to the choice of $\rho = \gamma = \frac{1}{\tau}$ in the penalty function (3) and are natural in the sense of Proposition 1. For the proximal steps in the parameters $\theta$ in (11) we have introduced a step size $\tau_\theta$ which – as we will see in Proposition 2 below – changes the descent metric opposed to $\tau$ which rather rescales the magnitude of the update.

We refer to one sweep of updates according to equations (7) - (12) as *ProxProp*, as it closely resembles the classical backpropagation (BackProp), but replaces the parameter update by a proximal mapping instead of an explicit gradient descent step. In the following subsection we analyze the convergence properties of ProxProp more closely.

### 4.2.1 CONVERGENCE OF PROXPROP

ProxProp inherits all convergence-relevant properties from the classical backpropagation algorithm, despite replacing explicit gradient steps with proximal steps: It minimizes the original network energy $J(\theta; X, y)$ as its fixed-points are stationary points of $J(\theta; X, y)$, and the update direction $\theta^{k+1} - \theta^k$ is a descent direction such that it converges when combined with a suitable optimizer. In particular, it is straight forward to combine ProxProp with popular optimizers such as Nesterov's accelerated gradient descent (Nesterov, 1983) or Adam (Kingma & Ba, 2015).

In the following, we give a detailed analysis of these properties.

**Proposition 2.** *For $l = 1, \ldots, L - 2$, the update direction $\theta^{k+1} - \theta^k$ computed by ProxProp meets*

$$\theta_l^{k+1} - \theta_l^k = -\tau \left( \frac{1}{\tau_\theta} I + (\nabla\phi(\cdot, a_{l-1}^k))(\nabla\phi(\cdot, a_{l-1}^k))^* \right)^{-1} \nabla_{\theta_l} J(\theta^k; X, y). \qquad (13)$$

In other words, ProxProp multiplies the gradient $\nabla_{\theta_l} J$ with the inverse of the positive definite, symmetric matrix

$$M_l^k := \frac{1}{\tau_\theta} I + (\nabla\phi(\cdot, a_{l-1}^k))(\nabla\phi(\cdot, a_{l-1}^k))^*, \qquad (14)$$

which depends on the activations $a_{l-1}^k$ of the forward pass. Proposition 2 has some important implications:

**Proposition 3.** *For any choice of $\tau > 0$ and $\tau_\theta > 0$, fixed points $\theta^*$ of ProxProp are stationary points of the original energy $J(\theta; X, y)$.*

Moreover, we can conclude convergence in the following sense.

**Proposition 4.** *The ProxProp direction $\theta^{k+1} - \theta^k$ is a descent direction. Moreover, under the (weak) assumption that the activations $a_l^k$ remain bounded, the angle $\alpha^k$ between $-\nabla_\theta J(\theta^k; X, y)$ and $\theta^{k+1} - \theta^k$ remains uniformly bounded away from $\pi/2$, i.e.*

$$\cos(\alpha^k) > c \geq 0, \qquad \forall k \geq 0, \qquad (15)$$

*for some constant $c$.*

Proposition 4 immediately implies convergence of a whole class of algorithms that depend only on a provided descent direction. We refer the reader to (Nocedal & Wright, 2006, Chapter 3.2) for examples and more details.

Furthermore, Proposition 4 states convergence for any minimization scheme in step ⓒ of Algorithm 2 that induces a descent direction in parameter space and thus provides the theoretical basis for a wide range of neural network optimization algorithms.

Considering the advantages of proximal steps over gradient steps, it is tempting to also update the auxiliary variables $a$ and $z$ in an implicit fashion. This corresponds to a proximal step in ⓑ of Algorithm 2. However, one cannot expect an analogue version of Proposition 3 to hold anymore. For example, if the update of $a_{L-2}$ in (8) is replaced by a proximal step, the propagated error does not correspond to the gradient of the loss function $\mathcal{L}_y$, but to the gradient of its Moreau envelope. Consequently, one would then minimize a different energy. While in principle this could result in an optimization algorithm with, for example, favorable generalization properties, we focus on minimizing the original network energy in this work and therefore do not further pursue the idea of implicit steps on $a$ and $z$.

### 4.2.2 Inexact solution of proximal steps

As we can see in Proposition 2, the ProxProp updates differ from vanilla gradient descent by the variable metric induced by the matrices $(M_l^k)^{-1}$ with $M_l^k$ defined in (14). Computing the ProxProp update direction $v_l^k := \frac{1}{\tau}(\theta_l^{k+1} - \theta_l^k)$ therefore reduces to solving the linear equation

$$M_l^k v_l^k = -\nabla_{\theta_l} J(\boldsymbol{\theta}^k; X, y), \tag{16}$$

which requires an efficient implementation. We propose to use a conjugate gradient (CG) method, not only because it is one of the most efficient methods for iteratively solving linear systems in general, but also because it can be implemented *matrix-free*: It merely requires the application of the linear operator $M_l^k$ which consists of the identity and an application of $(\nabla\phi(\cdot, a_{l-1}^k))(\nabla\phi(\cdot, a_{l-1}^k))^*$. The latter, however, is preimplemented for many linear transfer functions $\phi$ in common deep learning frameworks, because $\phi(x, a_{l-1}^k) = (\nabla\phi(\cdot, a_{l-1}^k))^*(x)$ is nothing but a forward-pass in $\phi$, and $\phi^*(z, a_{l-1}^k) = (\nabla\phi(\cdot, a_{l-1}^k))(z)$ provides the gradient with respect to the parameters $\theta$ if $z$ is the backpropagated gradient up to that layer. Therefore, a CG solver is straight-forward to implement in any deep learning framework using the existing, highly efficient and high level implementations of $\phi$ and $\phi^*$. For a fully connected network $\phi$ is a matrix multiplication and for a convolutional network the convolution operation.

As we will analyze in more detail in Section 5.1, we approximate the solution to (16) with a few CG iterations, as the advantage of highly precise solutions does not justify the additional computational effort in obtaining them. Using any number of iterations provably does not harm the convergence properties of ProxProp:

**Proposition 5.** *The direction $\tilde{v}_l^k$ one obtains from approximating the solution $v_l^k$ of (16) with the CG method remains a descent direction for any number of iterations.*

### 4.2.3 Convergence in the stochastic setting

While the above analysis considers only the full batch setting, we remark that convergence of Prox-Prop can also be guaranteed in the stochastic setting under mild assumptions. Assuming that the activations $a_l^k$ remain bounded (as in Proposition 4), the eigenvalues of $(M_l^k)^{-1}$ are uniformly contained in the interval $[\lambda, \tau_\theta]$ for some fixed $\lambda > 0$. Therefore, our ProxProp updates fulfill Assumption 4.3 in (Bottou et al., 2016), presuming the classic stochastic gradient fulfills them. This guarantees convergence of stochastic ProxProp updates in the sense of (Bottou et al., 2016, Theorem 4.9), i.e. for a suitable sequence of diminishing step sizes.

## 5 Numerical evaluation

ProxProp generally fits well with the API provided by modern deep learning frameworks, since it can be implemented as a network layer with a custom backward pass for the proximal mapping. We chose PyTorch for our implementation[1]. In particular, our implementation can use the API's GPU compute capabilities; all numerical experiments reported below were conducted on an NVIDIA Titan X GPU. To directly compare the algorithms, we used our own layer for either proximal or gradient update steps (cf. step Ⓒ in Algorithms 1 and 2). A ProxProp layer can be seamlessly integrated in a larger network architecture, also with other parametrized layers such as BatchNormalization.

### 5.1 Exact and approximate solutions to proximal steps

We study the behavior of ProxProp in comparison to classical BackProp for a supervised visual learning problem on the CIFAR-10 dataset. We train a fully connected network with architecture $3072 - 4000 - 1000 - 4000 - 10$ and ReLU nonlinearities. As the loss function, we chose the cross-entropy between the probability distribution obtained by a softmax nonlinearity and the ground-truth labels. We used a subset of 45000 images for training while keeping 5000 images as a validation set. We initialized the parameters $\theta_l$ uniformly in $[-1/\sqrt{n_{l-1}}, 1/\sqrt{n_{l-1}}]$, the default initialization of PyTorch.

---

[1]https://github.com/tfrerix/proxprop

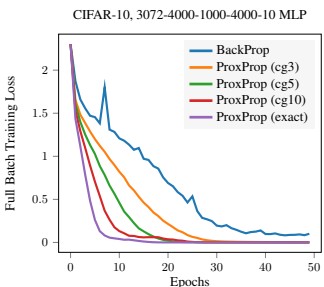 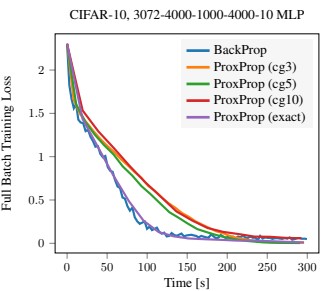 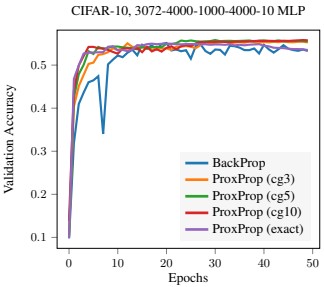

Figure 2: Exact and inexact solvers for ProxProp compared with BackProp. **Left:** A more precise solution of the proximal subproblem leads to overall faster convergence, while even a very inexact solution (only 3 CG iterations) already outperforms classical backpropagation. **Center & Right:** While the run time is comparable between the methods, the proposed ProxProp updates have better generalization performance ($\approx 54\%$ for BackProp and $\approx 56\%$ for ours on the test set).

Figure 2 shows the decay of the full batch training loss over epochs (left) and training time (middle) for a Nesterov momentum[2] based optimizer using a momentum of $\mu = 0.95$ and minibatches of size 500. We used $\tau_\theta = 0.05$ for the ProxProp variants along with $\tau = 1$. For BackProp we chose $\tau = 0.05$ as the optimal value we found in a grid search.

As we can see in Figure 2, using implicit steps indeed improves the optimization progress per epoch. Thanks to powerful linear algebra methods on the GPU, the exact ProxProp solution is competitive with BackProp even in terms of runtime.

The advantage of the CG-based approximations, however, is that they generalize to arbitrary linear transfer functions in a matrix-free manner, i.e. they are independent of whether the matrices $M_l^k$ can be formed efficiently. Moreover, the validation accuracies (right plot in Figure 2) suggest that these approximations have generalization advantages in comparison to BackProp as well as the exact ProxProp method. Finally, we found the exact solution to be significantly more sensitive to changes of $\tau_\theta$ than its CG-based approximations. We therefore focus on the CG-based variants of ProxProp in the following. In particular, we can eliminate the hyperparameter $\tau_\theta$ and consistently chose $\tau_\theta = 1$ for the rest of this paper, while one can in principle perform a hyperparameter search just as for the learning rate $\tau$. Consequently, there are no additional parameters compared with BackProp.

## 5.2 STABILITY FOR LARGER STEP SIZES

We compare the behavior of ProxProp and BackProp for different step sizes. Table 1 shows the final full batch training loss after 50 epochs with various $\tau$. The ProxProp based approaches remain stable over a significantly larger range of $\tau$. Even more importantly, deviating from the optimal step size $\tau$ by one order of magnitude resulted in a divergent algorithm for classical BackProp, but still provides reasonable training results for ProxProp with 3 or 5 CG iterations. These results are in accordance with our motivation developed in Section 4.1. From a practical point of view, this eases hyperparameter search over $\tau$.

| $\tau$ | 50 | 10 | 5 | 1 | 0.5 | 0.1 | 0.05 | $5 \cdot 10^{-3}$ | $5 \cdot 10^{-4}$ |
|---|---|---|---|---|---|---|---|---|---|
| BackProp | – | – | – | – | – | 0.524 | 0.091 | 0.637 | 1.531 |
| ProxProp (cg1) | 77.9 | 0.079 | 0.145 | 0.667 | 0.991 | 1.481 | 1.593 | 1.881 | 2.184 |
| ProxProp (cg3) | 94.7 | 0.644 | 0.031 | $2 \cdot 10^{-3}$ | 0.012 | 1.029 | 1.334 | 1.814 | 2.175 |
| ProxProp (cg5) | 66.5 | 0.190 | 0.027 | $3 \cdot 10^{-4}$ | $2 \cdot 10^{-3}$ | 0.399 | 1.049 | 1.765 | 2.175 |

Table 1: Full batch loss for conjugate gradient versions of ProxProp and BackProp after training for 50 epochs, where "–" indicates that the algorithm diverged to `NaN`. The implicit ProxProp algorithms remain stable for a significantly wider range of step sizes.

---

[2]PyTorch's Nesterov SGD for direction $d(\theta^k)$: $m^{k+1} = \mu m^k + d(\theta^k)$, $\theta^{k+1} = \theta^k - \tau(\mu m^{k+1} + d(\theta^k))$.

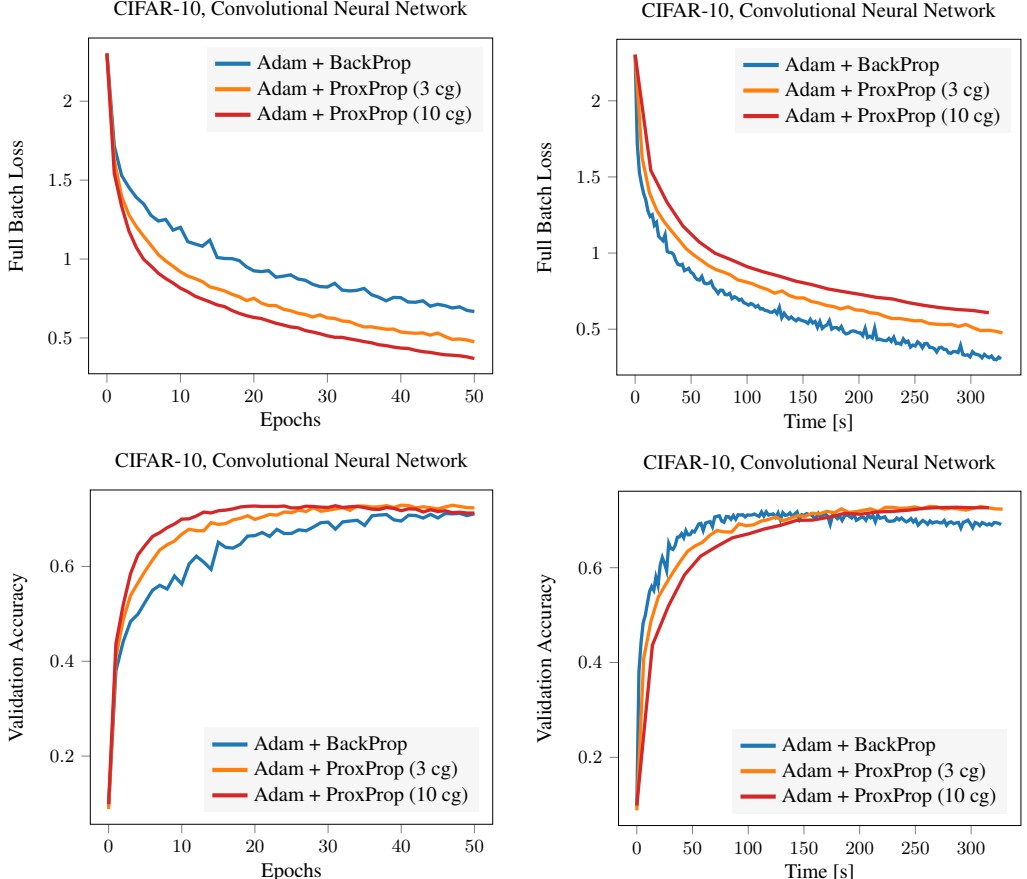

Figure 3: ProxProp as a first-order oracle in combination with the Adam optimizer. The proposed method leads to faster decrease of the full batch loss in epochs and to an overall higher accuracy on the validation set. The plots on the right hand side show data for a fixed runtime, which corresponds to a varying number of epochs for the different optimizers.

### 5.3   PROXPROP AS A FIRST-ORDER ORACLE

We show that ProxProp can be used as a gradient oracle for first-order optimization algorithms. In this section, we consider Adam (Kingma & Ba, 2015). Furthermore, to demonstrate our algorithm on a generic architecture with layers commonly used in practice, we trained on a convolutional neural network of the form:

$$\text{Conv}[16 \times 32 \times 32] \rightarrow \text{ReLU} \rightarrow \text{Pool}[16 \times 16 \times 16] \rightarrow \text{Conv}[20 \times 16 \times 16] \rightarrow \text{ReLU}$$
$$\rightarrow \text{Pool}[20 \times 8 \times 8] \rightarrow \text{Conv}[20 \times 8 \times 8] \rightarrow \text{ReLU} \rightarrow \text{Pool}[20 \times 4 \times 4] \rightarrow \text{FC} + \text{Softmax}[10 \times 1 \times 1]$$

Here, the first dimension denotes the respective number of filters with kernel size $5 \times 5$ and max pooling downsamples its input by a factor of two. We set the step size $\tau = 10^{-3}$ for both BackProp and ProxProp.

The results are shown in Fig. 3. Using parameter update directions induced by ProxProp within Adam leads to a significantly faster decrease of the full batch training loss in epochs. While the running time is higher than the highly optimized backpropagation method, we expect that it can be improved through further engineering efforts. We deduce from Fig. 3 that the best validation accuracy (72.9%) of the proposed method is higher than the one obtained with classical backpropagation (71.7%). Such a positive effect of proximal smoothing on the generalization capabilities of deep networks is consistent with the observations of Chaudhari et al. (2017b). Finally, the accuracies on the test set after 50 epochs are 70.7% for ProxProp and 69.6% for BackProp which suggests that the proposed algorithm can lead to better generalization.

## 6 Conclusion

We have proposed proximal backpropagation (ProxProp) as an effective method for training neural networks. To this end, we first showed the equivalence of the classical backpropagation algorithm with an algorithm that alternates between sequential gradient steps on a quadratic penalty function and forward passes through the network. Subsequently, we developed a generalization of Back-Prop, which replaces explicit gradient steps with implicit (proximal) steps, and proved that such a scheme yields a descent direction, even if the implicit steps are approximated by conjugate gradient iterations. Our numerical analysis demonstrates that ProxProp is stable across various choices of step sizes and shows promising results when compared with common stochastic gradient descent optimizers.

We believe that the interpretation of error backpropagation as the alternation between forward passes and sequential minimization steps on a penalty functional provides a theoretical basis for the development of further learning algorithms.

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

## APPENDIX

## A  THEORETICAL RESULTS

*Proof of Proposition 1.* We first take a gradient step on

$$E(\boldsymbol{\theta}, \boldsymbol{a}, \boldsymbol{z}) = \mathcal{L}_y(\phi(\theta_{L-1}, a_{L-2})) + \sum_{l=1}^{L-2} \frac{\gamma}{2} \|\sigma(z_l) - a_l\|^2 + \frac{\rho}{2} \|\phi(\theta_l, a_{l-1}) - z_l\|^2, \qquad (17)$$

with respect to $(\theta_{L-1}, a_{L-2})$. The gradient step with respect to $\theta_{L-1}$ is the same as in the gradient descent update,

$$\boldsymbol{\theta}^{k+1} = \boldsymbol{\theta}^k - \tau \nabla J(\boldsymbol{\theta}^k; X, y), \qquad (18)$$

since $J$ depends on $\theta_{L-1}$ only via $\mathcal{L}_y \circ \phi$.

The gradient descent step on $a_{L-2}$ in ⓐ yields

$$a_{L-2}^{k+1/2} = a_{L-2}^k - \tau \nabla_a \phi(\theta_{L-1}^k, a_{L-2}^k) \cdot \nabla_\phi \mathcal{L}_y(\phi(\theta_{L-1}^k, a_{L-2}^k)), \qquad (19)$$

where we use $a_{L-2}^{k+1/2}$ to denote the updated variable $a_{L-2}$ before the forward pass of the next iteration. To keep the presentation as clear as possible we slightly abused the notation of a right multiplication with $\nabla_a \phi(\theta_{L-1}^k, a_{L-2}^k)$: While this notation is exact in the case of fully connected layers, it represents the application of the corresponding linear operator in the more general case, e.g. for convolutions.

For all layers $l \leq L - 2$ note that due to the forward pass in Algorithm 1 we have

$$\sigma(z_l^k) = a_l^k, \quad \phi(\theta_l^k, a_{l-1}^k) = z_l^k \qquad (20)$$

and we therefore get the following update equations in the gradient step ⓑ

$$z_l^{k+1/2} = z_l^k - \tau\gamma\nabla\sigma(z_l^k)\left(\sigma(z_l^k) - a_l^{k+1/2}\right) = z_l^k - \nabla\sigma(z_l^k)\left(a_l^k - a_l^{k+1/2}\right), \qquad (21)$$

and in the gradient step ⓒ w.r.t. $a_{l-1}$,

$$\begin{aligned} a_{l-1}^{k+1/2} &= a_{l-1}^k - \tau\rho\nabla_a\phi(\theta_l^k, a_{l-1}^k) \cdot \left(\phi(\theta_l^k, a_{l-1}^k) - z_l^{k+1/2}\right) \\ &= a_{l-1}^k - \nabla_a\phi(\theta_l^k, a_{l-1}^k) \cdot \left(z_l^k - z_l^{k+1/2}\right). \end{aligned} \qquad (22)$$

Equations (21) and (22) can be combined to obtain:

$$z_l^k - z_l^{k+1/2} = \nabla\sigma(z_l^k)\nabla_a\phi(\theta_{l+1}^k, a_l^k) \cdot \left(z_{l+1}^k - z_{l+1}^{k+1/2}\right). \qquad (23)$$

The above formula allows us to backtrack the differences of the old $z_l^k$ and the updated $z_l^{k+1/2}$ up to layer $L - 2$, where we can use equations (21) and (19) to relate the difference to the loss. Altogether, we obtain

$$z_l^k - z_l^{k+1/2} = \tau \left(\prod_{q=l}^{L-2} \nabla\sigma(z_q^k)\nabla_a\phi(\theta_{q+1}^k, a_q^k)\right) \cdot \nabla_\phi \mathcal{L}_y(\phi(\theta_{L-1}^k, a_{L-2}^k)). \qquad (24)$$

By inserting (24) into the gradient descent update equation with respect to $\theta_l$ in ⓒ ,

$$\theta^{k+1} = \theta^k - \nabla_\theta\phi(\theta_l^k, a_{l-1}^k) \cdot \left(z_l^k - z_l^{k+1/2}\right), \qquad (25)$$

we obtain the chain rule for update (18). □

*Proof of Proposition 2.* Since only the updates for $\theta_l$, $l = 1, \dots, L - 2$, are performed implicitly, one can replicate the proof of Proposition 1 exactly up to equation (24). Let us denote the right hand side of (24) by $g_l^k$, i.e. $z_l^{k+1/2} = z_l^k - g_l^k$ and note that

$$\tau\nabla_{\theta_l} J(\boldsymbol{\theta}^k; X, y) = \nabla_\theta\phi(\cdot, a_{l-1}^k) \cdot g_l^k \qquad (26)$$

holds by the chain rule (as seen in (25)). We have eliminated the dependence of $\nabla_\theta \phi(\theta_l^k, a_{l-1}^k)$ on $\theta_l^k$ and wrote $\nabla_\theta \phi(\cdot, a_{l-1}^k)$ instead, because we assume $\phi$ to be linear in $\theta$ such that $\nabla_\theta \phi$ does not depend on the point $\theta$ where the gradient is evaluated anymore.

We now rewrite the ProxProp update equation of the parameters $\theta$ as follows

$$
\begin{aligned}
\theta_l^{k+1} &= \operatorname*{argmin}_\theta \; \frac{1}{2}||\phi(\theta, a_{l-1}^k) - z_l^{k+1/2}||^2 + \frac{1}{2\tau_\theta}||\theta - \theta_l^k||^2 \\
&= \operatorname*{argmin}_\theta \; \frac{1}{2}||\phi(\theta, a_{l-1}^k) - (z_l^k - g_l^k)||^2 + \frac{1}{2\tau_\theta}||\theta - \theta_l^k||^2 \\
&= \operatorname*{argmin}_\theta \; \frac{1}{2}||\phi(\theta, a_{l-1}^k) - (\phi(\theta^k, a_{l-1}^k) - g_l^k)||^2 + \frac{1}{2\tau_\theta}||\theta - \theta_l^k||^2 \\
&= \operatorname*{argmin}_\theta \; \frac{1}{2}||\phi(\theta - \theta^k, a_{l-1}^k) + g_l^k||^2 + \frac{1}{2\tau_\theta}||\theta - \theta_l^k||^2,
\end{aligned}
\tag{27}
$$

where we have used that $\phi$ is linear in $\theta$. The optimality condition yields

$$
0 = \nabla\phi(\cdot, a_{l-1}^k)(\phi(\theta_l^{k+1} - \theta^k, a_{l-1}^k) + g_l^k) + \frac{1}{\tau_\theta}(\theta_l^{k+1} - \theta_l^k)
\tag{28}
$$

Again, due to the linearity of $\phi$ in $\theta$, one has

$$
\phi(\theta, a_{l-1}^k) = (\nabla\phi(\cdot, a_{l-1}^k))^*(\theta),
\tag{29}
$$

where $^*$, denotes the adjoint of a linear operator. We conclude

$$
0 = \nabla\phi(\cdot, a_{l-1}^k)(\nabla\phi(\cdot, a_{l-1}^k))^*(\theta_l^{k+1} - \theta^k) + \nabla\phi(\cdot, a_{l-1}^k)g_l^k + \frac{1}{\tau_\theta}(\theta_l^{k+1} - \theta_l^k),
$$

$$
\Rightarrow \left(\frac{1}{\tau_\theta}I + \nabla\phi(\cdot, a_{l-1}^k)(\nabla\phi(\cdot, a_{l-1}^k))^*\right)(\theta_l^{k+1} - \theta_l^k) = -\nabla\phi(\cdot, a_{l-1}^k)g_l^k = -\tau\nabla_{\theta_l}J(\boldsymbol{\theta}^k; X, y),
\tag{30}
$$

which yields the assertion. $\qquad\square$

*Proof of Proposition 3.* Under the assumption that $\theta^k$ converges, $\theta^k \to \hat{\theta}$, one finds that $a_l^k \to \hat{a}_l$ and $z_l^k \to \hat{z}_l = \phi(\hat{\theta}_l, \hat{a}_{l-1})$ converge to the respective activations of the parameters $\hat{\theta}$ due to the forward pass and the continuity of the network. As we assume $J(\cdot; X, y)$ to be continuously differentiable, we deduce from (30) that $\lim_{k\to\infty} \nabla_{\theta_l} J(\boldsymbol{\theta}^k; X, y) = 0$ for all $l = 1, \dots, L - 2$. The parameters of the last layer $\theta_{L-1}$ are treated explicitly anyways, such that the above equation also holds for $l = L - 1$, which then yields the assertion. $\qquad\square$

*Proof of Proposition 4.* As the matrices

$$
M_l^k := \frac{1}{\tau_\theta}I + (\nabla\phi(\cdot, a_{l-1}^k))(\nabla\phi(\cdot, a_{l-1}^k))^*
\tag{31}
$$

(with the convention $M_{L-1}^k = I$) are positive definite, so are their inverses, and the claim that $\boldsymbol{\theta}^{k+1} - \boldsymbol{\theta}^k$ is a descent direction is immediate,

$$
\langle\theta_l^{k+1} - \theta_l^k, -\nabla_{\theta_l}J(\boldsymbol{\theta}^k; Y, x)\rangle = \tau\langle(M_l^k)^{-1}\nabla_{\theta_l}J(\boldsymbol{\theta}^k; Y, x), \nabla_{\theta_l}J(\boldsymbol{\theta}^k; Y, x)\rangle.
\tag{32}
$$

We still have to guarantee that this update direction does not become orthogonal to the gradient in the limit $k \to \infty$. The largest eigenvalue of $(M_l^k)^{-1}$ is bounded from above by $\tau_\theta$. If the $a_{l-1}^k$ remain bounded, then so does $\nabla\phi(\cdot, a_{l-1}^k)$ and the largest eigenvalue of $\nabla\phi(\cdot, a_{l-1}^k)\nabla\phi(\cdot, a_{l-1}^k)^*$ must be bounded by some constant $\tilde{c}$. Therefore, the smallest eigenvalue of $(M_l^k)^{-1}$ must remain bounded from from below by $(\frac{1}{\tau_\theta} + \tilde{c})^{-1}$. Abbreviating $v = \nabla_{\theta_l}J(\boldsymbol{\theta}^k; Y, x)$, it follows that

$$
\begin{aligned}
\cos(\alpha^k) &= \frac{\tau\langle(M_l^k)^{-1}v, v\rangle}{\tau||(M_l^k)^{-1}v||\,||v||} \\
&\geq \frac{\lambda_{\min}((M_l^k)^{-1})||v||^2}{||(M_l^k)^{-1}v||\,||v||} \\
&\geq \frac{\lambda_{\min}((M_l^k)^{-1})}{\lambda_{\max}((M_l^k)^{-1})}
\end{aligned}
\tag{33}
$$

which yields the assertion. □

*Proof of Proposition 5.* According to (Nocedal & Wright, 2006, p. 109, Thm. 5.3) and (Nocedal & Wright, 2006, p. 106, Thm. 5.2) the $k$-th iteration $x_k$ of the CG method for solving a linear system $Ax = b$ with starting point $x_0 = 0$ meets

$$x_k = \arg \min_{x \in \text{span}(b, Ab, \dots, A^{k-1}b)} \frac{1}{2} \langle x, Ax \rangle - \langle b, x \rangle, \tag{34}$$

i.e. is optimizing over an order-$k$ Krylov subspace. The starting point $x_0 = 0$ can be chosen without loss of generality. Suppose the starting point is $\tilde{x}_0 \neq 0$, then one can optimize the variable $x = \tilde{x} - \tilde{x}_0$ with a starting point $x_0 = 0$ and $b = \tilde{b} + A\tilde{x}_0$.

We will assume that the CG iteration has not converged yet as the claim for a fully converged CG iteration immediately follow from Proposition 4. Writing the vectors $b, Ab, \dots, A^{k-1}b$ as columns of a matrix $\mathcal{K}_k$, the condition $x \in \text{span}(b, Ab, \dots, A^{k-1}b)$ can equivalently be expressed as $x = \mathcal{K}_k \alpha$ for some $\alpha \in \mathbb{R}^k$. In terms of $\alpha$ our minimization problem becomes

$$x_k = \mathcal{K}_k \alpha = \arg \min_{\alpha \in \mathbb{R}^k} \frac{1}{2} \langle \alpha, (\mathcal{K}_k)^T A \mathcal{K}_k \alpha \rangle - \langle (\mathcal{K}_k)^T b, \alpha \rangle, \tag{35}$$

leading to the optimality condition

$$0 = (\mathcal{K}_k)^T A \mathcal{K}_k \alpha - (\mathcal{K}_k)^T b,$$
$$\Rightarrow x_k = \mathcal{K}_k ((\mathcal{K}_k)^T A \mathcal{K}_k)^{-1} (\mathcal{K}_k)^T b. \tag{36}$$

Note that $A$ is symmetric positive definite and can therefore be written as $\sqrt{A}^T \sqrt{A}$, leading to

$$(\mathcal{K}_k)^T A \mathcal{K}_k = (\sqrt{A} \mathcal{K}_k)^T (\sqrt{A} \mathcal{K}_k) \tag{37}$$

being symmetric positive definite. Hence, the matrix $((\mathcal{K}_k)^T A \mathcal{K}_k)^{-1}$ is positive definite, too, and

$$\langle x_k, b \rangle = \langle \mathcal{K}_k ((\mathcal{K}_k)^T A \mathcal{K}_k)^{-1} (\mathcal{K}_k)^T b, b \rangle$$
$$= \langle ((\mathcal{K}_k)^T A \mathcal{K}_k)^{-1} (\mathcal{K}_k)^T b, (\mathcal{K}_k)^T b \rangle > 0. \tag{38}$$

Note that $(\mathcal{K}_k)^T b$ is nonzero if $b$ is nonzero, as $\|b\|^2$ is its first entry.

To translate the general analysis of the CG iteration to our specific case, using any number of CG iterations we find that an approximate solution $\tilde{v}_l^k$ of

$$M_l^k v_l^k = -\nabla_{\theta_l} J(\boldsymbol{\theta}^k; X, y) \tag{39}$$

leads to

$$\langle \tilde{v}_l^k, -\nabla_{\theta_l} J(\boldsymbol{\theta}^k; X, y) \rangle > 0,$$

i.e., to $\tilde{v}_l^k$ being a descent direction. □

## B   PROXIMAL OPERATOR FOR LINEAR TRANSFER FUNCTIONS

In order to update the parameters $\theta_l$ of the linear transfer function, we have to solve the problem (11),

$$\theta^{k+1} = \underset{\theta}{\text{argmin}} \frac{1}{2} \|\phi(\theta, a^k) - z^{k+1/2}\|^2 + \frac{1}{2\tau_\theta} \|\theta - \theta^k\|^2. \tag{40}$$

Since we assume that $\phi$ is linear in $\theta$ for a fixed $a^k$, there exists a matrix $A^k$ such that

$$\text{vec}(\theta^{k+1}) = \underset{\theta}{\text{argmin}} \frac{1}{2} \|A^k \text{vec}(\theta) - \text{vec}(z^{k+1/2})\|^2 + \frac{1}{2\tau_\theta} \|\text{vec}(\theta) - \text{vec}(\theta^k)\|^2, \tag{41}$$

and the optimality condition yields

$$\text{vec}(\theta^{k+1}) = (I + \tau_\theta (A^k)^T A^k)^{-1} (\text{vec}(\theta^k) + (A^k)^T \text{vec}(z^{k+1/2})). \tag{42}$$

In the main paper we sometimes use the more abstract but also more concise notion of $\nabla\phi(\cdot, a^k)$, which represents the linear operator

$$\nabla\phi(\cdot, a^k)(Y) = \text{vec}^{-1}((A^k)^T \text{vec}(Y)). \tag{43}$$

To also make the above more specific, consider the example of $\phi(\theta, a^k) = \theta a^k$. In this case the variable $\theta$ may remain in a matrix form and the solution of the proximal mapping becomes

$$\theta^{k+1} = \left( z^{k+1/2} (a^k)^\top + \frac{1}{\tau_\theta} \theta^k \right) \left( a^k (a^k)^\top + \frac{1}{\tau_\theta} I \right)^{-1}. \tag{44}$$

Since $a^k \in \mathbb{R}^{n \times N}$ for some layer size $n$ and batch size $N$, the size of the linear system is independent of the batch size.

