# OpenReview forum: "Proximal Backpropagation"
_ICLR.cc/2018/Conference — Accept (Poster)_

### Official Review · AnonReviewer1 · 2017-11-26
**No theoretical result for its stochastic variant and how to choose stepsize**

**Rating:** 5
**Confidence:** 4

**Review:**

This work proposes to replace the gradient step for updating the network parameters to a proximal step (implicit gradient) so that a large stepsize can be taken.  Then to make it fast, the implicit step is approximated using conjugate gradient method because the step is solving a quadratic problem.

The theoretical result of the ProxProp considers the full batch, and it can not be easily extended to the stochastic variant (mini-batch). The reason is that the gradient of proximal is evaluated at the future point, and different functions will have different future points. While for the explicit gradient, it is assessed at the current point, and it is an unbiased one.

In the numerical experiment, the parameter \tau_\theta is sensitive to the final solution. Therefore, how to choose this parameter is essential. Given a new dataset, how to determine it for a good performance.

In Fig 3. The full batch loss of Adam+ProxProp is higher than Adam+BackProp regarding time, which is different from Fig. 2. Also, the figure shows that the performance of Adam+BackProp is worst than Adam+ProxProp even though the training loss of Adam+BackProp is smaller that of Adam+ProxProp. Does it happen on this dataset only or it is the case for many datasets?

---

> ### Author Response · Authors · 2017-12-11
> **Reply to "No theoretical result for its stochastic variant and how to choose stepsize"**
>
> We agree that the theoretical results in our paper do not address the stochastic setting directly. Our results do, however, show that the proposed method yields the descent in a variable (uniformly bounded) metric, which allows to prove convergence even in a stochastic setting, if typical assumptions about the vanilla stochastic gradient are made. We refer for instance to the Assumptions 4.3 in  https://arxiv.org/pdf/1606.04838.pdf as well as to the subsequent convergence analysis therein (Theorem 4.9). The practical setup in which most deep learning algorithms are used is, however, quite different from the setting used for convergence analysis. For example, rectified linear units or max pooling functions are not differentiable. We therefore focused our attention on the numerical evaluation and were able to demonstrate convergence behavior comparable to the state-of-the-art in a stochastic setting for common neural network architectures. Despite the bias, our descent direction leads to faster optimization progress w.r.t. to epochs than the classical gradient and is competitive w.r.t. runtime.
> We have included a remark about the convergence theory in a stochastic setting based on the above reference in the revised version of our paper.
>
> Regarding the numerical experiments, we found that the convolutional neural network is not very sensitive to the choice of tau_theta; the fully-connected network is more sensitive. The parameter can be chosen with the same hyperparameter methods one might use to find a suitable learning rate, e.g. a grid search. Intuitively, the parameter interpolates between a gradient step (small tau_theta) and an exact layer-wise minimization step (large tau_theta).
> For a fair comparison to BackProp (i.e. same number of free hyperparameters), we set either tau or tau_theta fixed to 1, and only tuned one of them. We expect that by tuning both parameters, the performance of the proposed method can be further improved.
>
> Fig. 2 and Fig. 3 show experiments for different architectures on the same dataset (CIFAR-10) and the results cannot directly be compared. Furthermore, it is not unusual that a model with higher training error has a better validation accuracy, i.e. generalizes better. We do in general not expect a definitive relation between training and validation curves for different datasets/architectures.
>
> We hope that we could clarify the bias aspect of our algorithm in the stochastic setting and  kindly ask you to consider our additional comments in your rating. Please let us know if you have any further questions.

---

### Official Review · AnonReviewer3 · 2017-11-27
**Interesting idea, though not yet clear if it often leads to better results**

**Rating:** 7
**Confidence:** 4

**Review:**

The paper uses a lesser-known interpretation of the gradient step of a composite function (i.e., via reverse mode automatic differentiation or backpropagation), and then replaces one of the steps with a proximal step. The proximal step requries the solution of a positive-definite linear system, so it is approximated using a few iterations of CG. The paper provides theory to show that their proximal variant (even with the CG approximations) can lead to convergent algorithms (and since practical algorithms are not necessarily globally convergent, most of the theory shows that the proximal variant has similar guarantees to a standard gradient step).

On reading the abstract and knowing quite a bit about proximal methods, I was initially skeptical, but I think the authors have done a good job of making their case. It is a well-written, very clear paper, and it has a good understanding of the literature, and does not overstate the results. The experiments are serious, and done using standard state-of-the-art tools and architectures. Overall, it is an interesting idea, and due to the current focus on neural nets, it is of interest even though it is not yet providing substantial improvements.

The main drawback of this paper is that there is no theory to suggest the ProxProp algorithm has better worst-case convergence guarantees, and that the experiments do not show a consistent benefit (in terms of time) of the method. On the one hand, I somewhat agree with the authors that "while the running time is higher... we expect that it can be improved through further engineering efforts", but on the other hand, the idea of nested algorithms ("matrix-free" or "truncated Newton") always has this issue. A very similar type of ideas comes up in constrained or proximal quasi-Newton methods, and I have seen many papers (or paper submissions) on this style of method (e.g., see the 2017 SIAM Review paper on FWI by Metivier et al. at https://doi.org/10.1137/16M1093239). In every case, the answer seems to be that it can work on *some problems* and for a few well chosen parameters, so I don't yet buy that ProxProp is going to make a huge savings on a wide-range of problems.

In brief: quality is high, clarity is high, originality is high, and significance is medium.
Pros: interesting idea, relevant theory provided, high-quality experiments
Cons: no evidence that this is a "break-through" idea

Minor comments:

- Theorems seemed reasonable and I have no reason to doubt their accuracy

- No typos at all, which I find very unusual. Nice job!

- In Algo 1, it would help to be more explicit about the updates (a), (b), (c), e.g., for (a), give a reference to eq (8), and for (b), reference equations (9,10).  It's nice to have it very clear, since "gradient step" doesn't make it clear what the stepsize is, and if this is done in a "Jacob-like" or "Gauss-Seidel-like" fashion. (c) has no reference equation, does it?

- Similarly, for Algo 2, add references. In particular, tie in the stepsizes tau and tau_theta here.

- Motivation in section 4.1 was a bit iffy. A larger stepsize is not always better, and smaller is not worse. Minimizing a quadratic f(x) = .5||x||^2 will converge in one step with a step-size of 1 because this is well-conditioned; on the flip side, slow convergence comes from lack of strong convexity, or with strong convexity, ill-conditioning of the Hessian (like a stiff ODE).

- The form of equation (6) was very nice, and you could also point out the connection with backward Euler for finite-difference methods. This was the initial setting of analysis for most of original results that rely on the proximal operator (e.g., Lions and Mercier 1970s).

- Eq (9), this is done component-wise, i.e., Hadamard product, right?

- About eq (12), even if softmax cross-entropy doesn't have a closed-form prox (and check the tables of Combettes and Pesquet), because it is separable (if I understand correctly) then it ought to be amenable to solving with a handful of Newton iterations which would be quite cheap.

Prox tables (see also the new edition of Bauschke and Combettes' book): P. L. Combettes and J.-C. Pesquet, "Proximal splitting methods in signal processing," in: Fixed-Point Algorithms for Inverse Problems in Science and Engineering (2011) http://www4.ncsu.edu/~pcombet/prox.pdf

- Below prop 4, discussing why not to make step (b) proximal, this was a bit vague to me. It would be nice to expand this.

- Page 6 near the top, to apply the operator, in the fully-connected case, this is just a matrix multiply, right? and in a conv net, just a convolution? It would help the reader to be more explicit here.

- Section 5.1, 2nd paragraph, did you swap tau_theta and tau, or am I just confused? The wording here was confusing.

- Fig 2 was not that convincing since the figure with time showed that either usual BackProp or the exact ProxProp were best, so why care about the approximate ProxProp with a few CG iterations? The argument of better generalization is based on very limited experiments and without any explanation, so I find that a weak argument (and it just seems weird that inexact CG gives better generalization).  The right figure would be nice to see with time on the x-axis as well.

- Section 5.2, this was nice and contributed to my favorable opinion about the work. However, any kind of standard convergence theory for usual SGD requires the stepsize to change per iteration and decrease toward zero. I've heard of heuristics saying that a fixed stepsize is best and then you just make sure to stop the algorithm a bit early before it diverges or behaves wildly -- is that true here?

- Final section of 5.3, about the validation accuracy, and the accuracy on the test set after 50 epochs. I am confused why these are different numbers. Is it just because 50 epochs wasn't enough to reach convergence, while 300 seconds was? And why limit to 50 epochs then? Basically, what's the difference between the bottom two plots in Fig 3 (other than scaling the x-axis by time/epoch), and why does ProxProp achieve better accuracy only in the right figure?

---

> ### Author Response · Authors · 2017-12-11
> **Reply to "Interesting idea, though not yet clear if it often leads to better results"**
>
> We agree that the magnitude of the step size on its own does not determine the convergence speed. While we stated the largest eigenvalue of the Hessian in the CIFAR-10 data as an exemplary restriction of the explicit step size (e.g. in a one layer network), it is also true that the Hessian is very ill-conditioned: In fact, the smallest eigenvalue differs from the largest one by about 7 orders of magnitude! Similar to some stiff ODEs, we believe that implicit steps behave favorable in such a case. In particular, implicit steps never become unstable (on the respective (sub-)problem).
>
> In section 5.1 we indeed picked \tau=1 and tau_\theta = 0.05 as for fully-connected networks this worked better than fixing tau_\theta = 1 and tuning \tau. We have fixed one of the parameters to have the same number of hyperparameters as for BackProp and expect a tuning of both parameters to further lead to an improved performance.
>
> There are several motivations to consider an inexact solve with CG beyond the cost of the linear solve. For an exact solve one has to explicitly construct the problem matrix. This is readily available for fully-connected nets, but needs to be computed for convolutional layers, which might be costly in memory and compute time. Additionally, from a practical point of view one would like to leverage existing implementations without additional coding overhead. Both aspects can be exploited by providing the forward/backward operation of your favorite autodiff library as an abstract linear operator to the CG solver.
>
> For this paper, we concentrated working out the effect of explicit vs. implicit gradient steps, and did intentionally not mix dynamic step size effects with these observations.
>
> The validation accuracies are computed on the validation set, i.e., a set that is not considered for training, but used for tuning the hyperparameters. This is distinct from a held back test set on which we just computed the final accuracy. The bottom two plots in Fig. 4 indeed only differ by the scaling of the x-axis. However, since BackProp is faster than our ProxProp variants per iteration, the plot against time contains data for 300s training time for every method. Consequently, the lower right plot shows more than 50 epochs of the Adam + BackProp algorithm.
>
> Please let us know if you have any further questions.

---

### Official Review · AnonReviewer2 · 2017-11-28
**A good paper but which should compare with BFGS techniques**

**Rating:** 6
**Confidence:** 4

**Review:**

Summary:

Using a penalty formulation of backpropagation introduced in a paper of Carreira-Perpinan and Wang (2014), the current submission proposes to minimize this formulation using explicit step for the update of the variables corresponding to the backward pass, but implicit steps for the update of the parameters of the network. The implicit steps have the advantage that the choice of step-size is replaced by a choice of a proximity coefficient, which the advantage that while too large step-size can increase the objective, any value of the proximity coefficient yields a proximal mapping guaranteed to decrease the objective.
The implicit are potentially one order of magnitude more costly than an explicit step since they require
to solve a linear system, but can be solved (exactly or partially) using conjugate gradient steps. The experiments demonstrate that the proposed algorithm are competitive with standard backpropagation and potentially faster if code is optimized further. The experiments show also that in on of the considered case the generalization accuracy is better for the proposed method.

Summary of the review:

The paper is well written, clear, tackles an interesting problem.
But, given that the method is solving a formulation that leverages second order information, it would seem reasonable to compare with existing techniques that leverage second order information to learn neural networks, namely BFGS, which has been studied for deep learning (see the references to Li and Fukushima (2001) and Ngiam et al (2011) below).

Review:

Using an implicit step leads to a descent step in a direction which is different than the gradient step.
Based on the experiment, the step in the implicit direction seems to decrease faster the objective, but the paper does not make an attempt to explain why. The authors must nonetheless have some intuition about this. Is it because the method can be understood as some form of block-coordinate Newton with momentum? It would be nice to have an even informal explanation.

Since a sequence of similar linear systems have to be solved could a preconditioner be gradually be solved and updated from previous iterations, using for example a BFGS approximation of the Hessian or other similar technique. This could be a way to decrease the number of CG iterations that must done at each step. Or can this replaced by a single BFGS style step?

The proposed scheme is applicable to the batch setting when most deep network are learned using stochastic gradient type methods. What is the relevance/applicability of the method given this context?

In fact given that the proposed scheme applies in the batch case, it seems that other contenders that are very natural are applicable, including BFGS variants for the non-convex case (

see e.g. Li, D. H., & Fukushima, M. (2001). On the global convergence of the BFGS method for nonconvex unconstrained optimization problems. SIAM Journal on Optimization, 11(4), 1054-1064.

and

J. Ngiam, A. Coates, A. Lahiri, B. Prochnow, Q. V. Le, and A. Y. Ng,
“On optimization methods for deep learning,” in Proceedings of the 28th
International Conference on Machine Learning, 2011, pp. 265–272.

)

or even a variant of BFGS which makes a block-diagonal approximation to the Hessian with one block per layer. To apply BFGS, one might have to replace the RELU function by a smooth counterpart..

How should one choose tau_theta?

In the experiments the authors compare with classical backpropagation, but they do not compare with
the explicit step of Carreira-Perpinan and Wang? This might be a relevant comparison to add to establish more clearly that it is the implicit step that yields the improvement.





Typos or question related to notations, details etc:

In the description of algorithm 2: the pseudo-code does not specify that the implicit step is done with regularization coefficient tau_theta

In equation (10) is z_l=z_l^k or z_l^(k+1/2) (I assume the former).

6th line of 5.1 theta_l is initialised uniformly in an interval -> could you explain why and/or provide a reference motivating this ?

8th line of 5.1 you mention Nesterov momentum method -> a precise reference and precise equation to lift ambiguities might be helpful.

In section 5.2 the reference to Table 5.2 should be Table 1.

---

> ### Author Response · Authors · 2017-12-11
> **Reply to "A good paper but which should compare with BFGS techniques"**
>
> While you are right that Eq. (13) suggests that the algorithm leverages second-order information for the layer-wise(!) quadratic subproblem, ProxProp is much closer to classical backpropagation than to a (quasi-)Newton method of the whole network (the limit-case \tau_\theta -> 0, \tau -> \infty in the Equation in Prop. 2 recovers BackProp).
> The Hessian matrix consists of second-order derivatives, while our metric is purely formed of the forward pass activations. Consider for example a layer at depth L (somewhere in the middle of the network). Then the Hessian (and any decent approximation) would depend on the network components at depth larger than L (in particular the final loss). Consequently, this Hessian of the overall energy changes when components at depth larger than L are modified. However, our metric at layer L is not affected by this modification. Hence the approach is quite distinct from quasi-Newton methods such as BFGS.
> We have therefore focused on comparing our algorithm with first-order methods that are conceptually closer and are at the same time considered current state-of-the-art.
>
> Our intuition is that ProxProp decreases the energy faster due to the advantage of implicit minimization steps over explicit steps as discussed in section 4.1. If you consider eq. (13), then in this metric the layer-wise quadratic subproblem is better conditioned. Intuitively, in this metric it becomes easier to make the quadratic fit for the last term in eq. (3). Other lines of work are partially motivated by the same issue, e.g. the commonly used BatchNormalization.
>
> We did not further go into preconditioners for the linear system as very few CG iterations already suffice for a good solution. Note that we warmstart the CG solver and expect the solution to be ‘close’ because of the proximal term. We have focused on the conjugate gradient approximation (also because of its abstract implementation advantages), but considering other approximations to the quadratic subproblem could be an interesting direction.
>
> It is correct that our theorems address the full batch setting. Since our method, however, still yields a descent in a different (variable) metric, the techniques discussed in https://arxiv.org/pdf/1606.04838.pdf section 4.1 are applicable to extend the analysis even to a stochastic setting, if one may assume a sufficiently friendly energy. Since in practice common neural nets have non-smooth energies, we have focused our paper on numerically demonstrating results comparable to the state-of-the-art by evaluating the algorithm in a stochastic mini-batch setting. We have added a remark and the above reference about extending the convergence analysis to the revised version of our paper.
>
> In our experiments, tau_theta is an additional hyperparameter that can be chosen just as the learning rate, e.g. via a hyperparameter grid search. However, we noticed that performance of our convolutional network architecture is not very sensitive to tau_theta.
>
> As stated after eq. (3), the method by Carreira-Perpinan and Wang is very different from our approach and also from explicit steps on the penalty functional. They perform block-coordinate minimization steps and also do not perform any forward passes. Nevertheless, we experimented with the MATLAB implementation kindly provided by the authors, but found that the numerical performance is far from efficient implementations of current state-of-the-art optimizers. We therefore didn’t see additional value - conceptually and numerically - of including this comparison in our paper.
>
> We chose the standard weight initialization of PyTorch and intentionally did not further tune this hyperparameter to avoid confounding effects.
>
> We used the Nesterov momentum method as implemented in PyTorch (http://pytorch.org/docs/master/optim.html#torch.optim.SGD) and will add a remark.
>
> In conclusion, we hope that we could clarify why we did not compare with BFGS style methods. Given your otherwise very positive review, we would appreciate if you reconsidered your rating. Please let us know if you have any further questions.

---

> > ### Comment · AnonReviewer2 · 2018-01-12
> > **Comments after rebuttal**
> >
> > I am convinced by the arguments put forward to explain why there is no comparison with the method of Carreira-Perpinan and Wang reported in the paper. It might be relevant to comment on this in the paper however.
> >
> > I understand that the formulation could be quite different than using second order information, and my point was not to say that they are similar in terms of functions.
> > What I mean is that the cost of computing a descent step of the algorithm is comparable with the cost of some  quasi-Newton methods (that would e.g. use block diagonal approximation of the Hessian) and on that ground the comparison would seem relevant. However, one could argue that the most important baseline is considered in the paper.
> >
> > In conclusion the paper would be more compelling with more comparison and I agree with Reviewer 3 that the significance is therefore unclear, but the approach proposed remains sound and interesting.

---

### Author Response · Authors · 2017-12-11
**Replies and revised PDF considering reviewers' comments**

We thank the reviewers for their constructive feedback. We have posted individual replies below the reviewers' comments and have uploaded a revised version of the PDF. The changes are marked in blue color.

---

### Decision · Program_Chairs · 2018-01-29
**ICLR 2018 Conference Acceptance Decision**

**Decision:**

Accept (Poster)

**Comment:**

Pros:
+ Clear, well-written paper that tackles an interesting problem.
+ Interesting potential connections to other approaches in the literature such as Carreira-Perpiñán and Wang, 2014 and Taylor et al., 2016.
+ Paper shows good understanding of the literature, has serious experiments, and does not overstate the results.

Cons:
- Theory only addresses gradient descent, not stochastic gradient descent.
- Because the optimization process is similar to BFGS, it would make sense to have an empirical comparison against some second-order method, even though the proposed algorithm is more like standard backpropagation.

This paper is a nice first step in an interesting direction, and belongs in ICLR if there is sufficient space.